# "Even though I am alone, I feel that we are many" - An appreciative inquiry study of asynchronous, provider-to-provider teleconsultations in Turkana, Kenya

**M. Whitney Fry**[1¤a], **Salima Saidi**[1], **Abdirahman Musa**[2], **Vanessa Kithyoma**[1¤b], **Pratap Kumar**[1,3]*

1 Health-E-Net Limited, Nairobi, Kenya, 2 Ministry of Health Services & Sanitation, Turkana County, Kenya, 3 Institute of Healthcare Management, Strathmore University Business School, Nairobi, Kenya

¤a Current address: Iris Group, Nairobi, Kenya
¤b Current address: mHealth Kenya Limited, Nairobi, Kenya
* pkumar@strathmore.edu

**Data Availability Statement:** Interviews and focus group discussion guides, consent forms and analysis files) are available in the Qualitative Data

## Abstract

Non-physician clinicians (NPCs) in low and middle-income countries (LMICs) often have little physical proximity to the resources–equipment, supplies or skills–needed to deliver effective care, forcing them to refer patients to distant sites. Unlike equipment or supplies, which require dedicated supply chains, physician/specialist skills needed to support NPCs can be sourced and delivered through telecommunication technologies. In LMICs however, these skills are scarce and sparsely distributed, making it difficult to implement commonly used real-time (synchronous), hub-and-spoke telemedicine paradigms. An asynchronous teleconsultations service was implemented in Turkana County, Kenya, connecting NPCs with a volunteer network of remote physicians and specialists. In 2017–18, the service supported over 100 teleconsultations and referrals across 20 primary healthcare clinics and two hospitals. This qualitative study aimed to explore the impact of the telemedicine intervention on health system stakeholders, and perceived health-related benefits to patients. Data were collected using Appreciative Inquiry, a strengths-based, positive approach to assessing interventions and informing systems change. We highlight the impact of provider-to-provider asynchronous teleconsultations on multiple stakeholders and healthcare processes. Provider benefits include improved communication and team work, increased confidence and capacity to deliver services in remote sites, and professional satisfaction for both NPCs and remote physicians. Health system benefits include efficiency improvements through improved care coordination and avoiding unnecessary referrals, and increased equity and access to physician/specialist care by reducing geographical, financial and social barriers. Providers and health system managers recognised several non-health benefits to patients including increased trust and care seeking from NPCs, and social benefits of avoiding unnecessary referrals (reduced social disruption, displacement and costs). The findings reveal the wider impact that modern teleconsultation services enabled by mobile technologies and algorithms can have on LMIC communities and health systems. The study highlights the importance of viewing provider-to-provider teleconsultations as complex health

Repository database:https://doi.org/10.5064/
F6UURYON.

**Funding:** Funding was provided from DFID through
the County Innovation Challenge Fund managed by
Options Consulting Services. None of the authors
were or are affiliated with the funding organisation
or its affiliates. The funder provided support in the
form of salaries for authors SS, AM, VK and PK,
but neither the the funding organisation nor its
affiliates played any role in the study design, data
collection and analysis, decision to publish, or
preparation of the manuscript. The specific roles of
these authors are articulated in the "author
contributions" section.

**Competing interests:** I have read the journal's
policy and the authors of this manuscript have the
following competing interests: SS and PK are
currently affiliated to Health-E-Net Limited, which
has commercial interests in the technology
described in the manuscript. This does not alter
our adherence to PLOS ONE policies on sharing
data and materials.

service delivery interventions with multiple pathways and processes that can ultimately
improve health outcomes.

## Introduction

Telemedicine services, including teleconsultations, have the potential to overcome geographi-
cal, social and financial barriers to accessing healthcare–all major global health challenges [1].
The delivery of remote medical consultations, recognized today as teleconsultations, can be
traced back to medical advice provided through letters in 1700s Scotland [2]. Telemedicine has
since advanced through a variety of communication modalities, and today is predominantly
conducted over mobile phone and internet [3]. However, the promise that teleconsultations
offer more practical, affordable and sustainable solutions than traditional consultations, espe-
cially in low and middle-income country (LMIC) health systems, has yet to be demonstrated at
scale [4–6].

One challenge in incorporating teleconsultations into existing healthcare services is a focus
on real-time (or synchronous) services either through telephone, or more commonly, video-
conferencing. This paradigm depends on extensive technological capacity and requires multi-
ple healthcare professionals (and often, patients) in different locations to abide by the same
schedule [7]. Despite these logistical (and related financial) challenges, video-based real-time
teleconsultations are commonly used and evaluated [8]. The alternative, asynchronous (or
store-and-forward) telemedicine paradigm has seen most success in tele-radiology, including
in LMIC settings [9, 10]. However the use of asynchronous teleconsultations (ATCs) as an
alternative to face-to-face consultations has been limited, with the exception of tele-dermatol-
ogy [11–13]. Despite not allowing for immediate feedback and dialogue, ATCs have numerous
advantages over their real-time counterpart: the paradigm is relatively easy to integrate with
clinical workflows, allows high-resolution images and other medical data to be shared and
reviewed, and involves lower costs to configure and operate [7]. ATCs are likely more practical
in LMIC settings, which often have limited and low-bandwidth internet connectivity, and
unreliable electricity. While there is growing literature on the use of asynchronous telemedi-
cine to support healthcare in underserved areas, its application and impact varies [14–20],
highlighting the need for both a deeper understanding of the paradigm, and more studies of
interventions at scale [21–23].

ATCs provide a means to overcome one of the defining challenges of LMIC health sys-
tems–the severe shortages of physicians and specialists, especially in rural areas [5]. There are
two or fewer doctors per 10,000 people in 37 countries in Africa [24]. As a result, non-physi-
cian clinicians (NPCs)–nurses, nurse practitioners, clinical officers (COs)–deliver the majority
of primary healthcare services, as well as a growing share of specialized services like caesarean
sections, closed fracture care, etc. [25, 26]. While the scope of practice of each cadre of NPC
varies by country, the number of NPCs have been steadily increasing as LMICs address the
need for additional human resources to deliver healthcare services [27]. The formal role of
physicians in these health systems is shifting to involve supporting NPCs to deliver high-qual-
ity care in their increasing scope of services. Such support is already being provided via calls
and communication platforms like WhatsApp [28, 29], but these efforts and technologies are
not built with medical consultations, privacy and clinical workflows in mind [30].

Regardless of the technologies used, provider-to-provider teleconsultations are at the inter-
section of the evolving role of physicians and NPCs. Understanding this telemedicine

paradigm, its challenges, benefits and impact will guide the design of future LMIC health systems [20]. We therefore studied the impact of an ATC service implemented in Turkana, a remote, marginalised region in north-western Kenya [31]. The objective of this research was to assess strengths of ATCs, understand their contribution to different stakeholders in the health system, and examine how the benefits of ATCs might be sustained if the services are integrated into the Turkana County health system.

## Materials and methods

This study was approved by the Strathmore University Institutional Review Board (SU-IRB 0008/15). The study included human participants in interviews and focus group discussions. Written consent was obtained from all participants.

### Intervention

The 'networked referral management and clinical decision support' project (nREM) was implemented in Turkana County, Kenya, in 2016. The project enabled NPCs in 20 remote primary healthcare clinics (PHCs) and two secondary hospitals to request ATCs from a distributed network of physicians and specialists. NPCs were trained to electronically document and share non-emergency cases that they would typically attempt to refer to a higher-level facility (ambulances are frequently unavailable to support referral, and referral advice is often not adhered to by patients in LMIC settings like Turkana for various reasons). The ATC technology platform was custom built to support referral and teleconsultation workflows (Fig 1), and designed to be: a) usable, with minimal training on the technology itself (e.g. app download, installation, use), and b) usable on commonly available and used mobile devices in Kenya (including personal smartphones), by both NPCs and remote physicians. One Android tablet was provided to each nREM project site in Turkana.

A volunteer network of remote physicians, including general physicians (known as medical officers (MOs) in Kenya) and specialists, was recruited to support NPCs via ATCs to either manage cases locally or encourage referral. The network included current and former employees of the county government, non-governmental organisations (NGOs) operating in the county, and individuals with experience and interest in supporting healthcare in Turkana or similarly resource-constrained regions [32]. Between 2017 and 2018 the project trained and supported over 100 NPCs (81 nurses, 26 COs) and 11 MOs from 20 PHCs and two referral hospitals in Turkana County, and enabled over 100 ATCs and electronic referrals.

### Study design

This study evaluated the intervention using Appreciative Inquiry, a qualitative research approach. Considered both a method and a theory of change, Appreciative Inquiry is an iterative process that generates data on existing positive elements in a system, while also leveraging these data to accelerate positive change [33]. Appreciative Inquiry was selected because of its ability to highlight often-understated areas of significance in innovation research, specifically positive and transformative elements in a system that may be advantageous for sustained growth [34]. The research team applied the first three phases of the Appreciative Inquiry model in the study: *Discovery*, *Dream*, and *Design* (the Appreciative Inquiry 4-D model is represented in Fig 2). An application of the data was intended to complete the Appreciative Inquiry approach into its fourth phase, *Delivery*.

This article documents findings from the *Discovery* phase, a comprehensive construction of stakeholder and systems-level benefits of ATCs integration in Turkana County. Data from *Dream* and *Design* phases are not documented here. In the *Discovery* phase, respondents

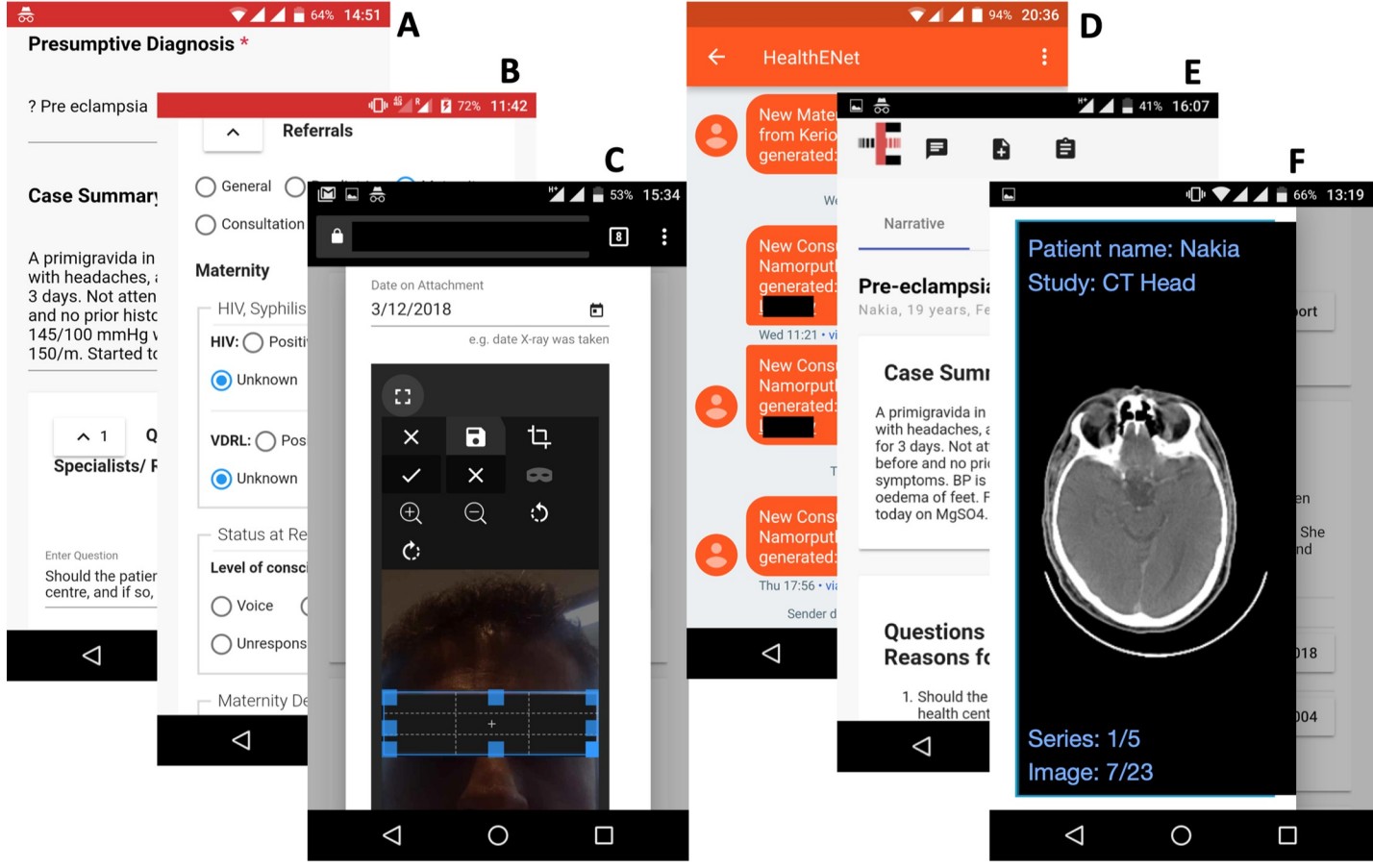

**Fig 1. Mobile phone screenshots of the ATC platform.** A. NPC interface on a mobile phone browser to enter case information in narrative form. B. NPC interface to enter structured data for selected information (e.g. HIV/VDRL status, ambulance use). C. NPC interface to capture images using the mobile device camera and redact private and confidential information. D. SMS notifications received by remote physicians including hyperlink to the case. E. Webpage of ATC/referral case accessed by the remote physician by clicking on the link in the SMS. The webpage includes chat, scheduling and reporting functionalities F. Example of a medical image in DICOM format viewed on the mobile browser by the remote physician (the NPC interface allows browser-based upload of DICOM data, if available). All data shown are exemplars created for illustration purposes. Reproduced under a CC BY license, with permission from Health-E-Net Limited, original copyright 2016".

described the best of what existed in the platform, using storytelling techniques to reflect on high-point experiences in their interactions with ATCs. In addition to primary data collection, the study team incorporated secondary quantitative data from the nREM project, in order to enhance findings in select content areas.

## Data collection

Primary data were collected in Nairobi, Kenya (with remote physicians) and in Lodwar, capital city of Turkana County, Kenya (with NPCs, MOs in the referral hospital, and County health system stakeholders) between February and June 2018. This was approximately one year after the initiation of ATCs in Turkana. Research participants (Table 1) were recruited through the nREM user network, with participation prioritised based on availability of engagement. Facilitation of key informant interviews (KIIs) and focus group discussions (FGDs) was conducted by a public health and Appreciative Inquiry specialist within the Health-E-Net team. All data collection followed the Appreciative Inquiry format, using semi-structured interview and FGD guides that were developed by the facilitator.

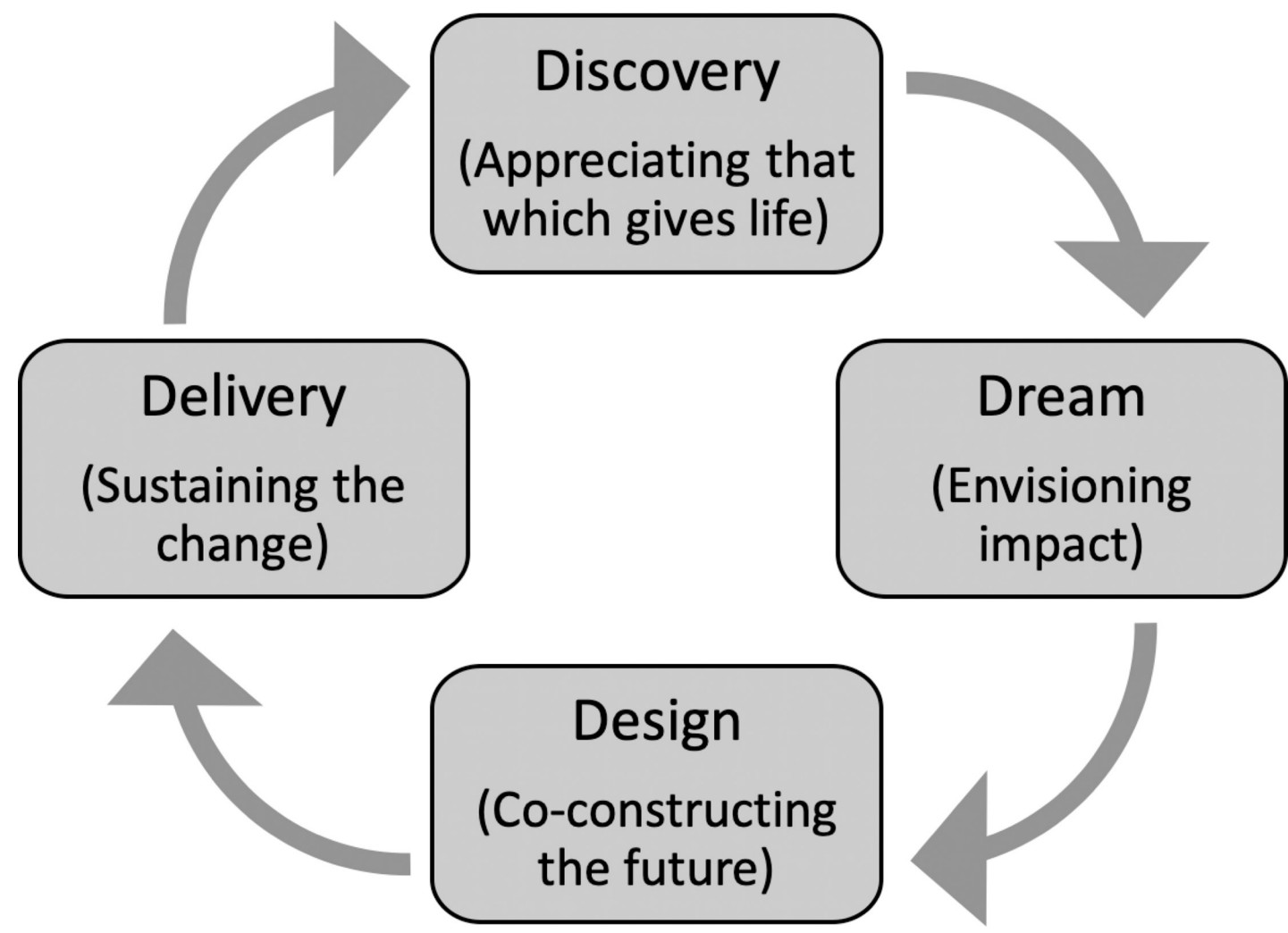

**Fig 2. The Appreciative Inquiry 4-D Model.** A schematic describing the four phases of Appreciative Inquiry [33]. The 'Discovery' phase aims to identify and appreciate what works; the 'Dream' phase involves imagining what might be; the 'Design' phase involves developing systems that leverage the best of what was and what might be; the 'Delivery' phase involves implementation or delivery of the proposed design.

**Table 1. Participants in the Appreciative Inquiry research.**

| Participant type | Selection criteria | Sample size; Method | Location |
|---|---|---|---|
| *Remote physicians* | Qualification: Medical Officer or Specialist Conducted at least two ATCs through nREM | n = 8 Key Informant Interviews | Nairobi, Kenya |
| *Non-Physician Clinicians* | Qualification: Registered nurse or Clinical Officer Based in a rural primary healthcare facility in Turkana Requested and received at least two ATCs through nREM | n = 12 Focus Group Discussions | Turkana, Kenya |
| *County Stakeholders* | Representatives of the following organizations:<br>• County Health Management Team<br>• Catholic Diocese of Lodwar (health NGO in Turkana)<br>• UNICEF (coordinating health services in Turkana) | n = 6 Key Informant Interviews | Turkana, Kenya |
| *Physicians in Referral Hospital* | Qualification: Medical Officers Based in Lodwar County Referral Hospital | n = 5 Focus Group Discussion | Turkana, Kenya |

List of participant types, their numbers, selection criteria and research methods used in the Appreciative Inquiry research.

Details and purpose of the research were shared with participants prior to their participation, and informed consent was received from all participants before initiating FGDs and KIIs. All sessions were audio recorded and transcribed verbatim, and transcription data were kept confidential for the duration of the data analysis process. Consent forms, KII and FGD guides, and the analysis codebook are accessible online in the qualitative data repository (https://doi.org/10.5064/F6UURYON).

## Analysis

Applied thematic analysis was used in the qualitative data analysis process, with individual respondents as the units of analysis [35]. Transcribed data were entered into *Dedoose* (v.6.1.18, 2014, Los Angeles, CA: SocioCultural Research Consultants, LLC; www.dedoose.com) for data management, excerpting, coding, and analysis. Definitions for analysis were outlined in the study's codebook to facilitate coding consistency, which was conducted solely by the lead researcher. Primary codes were descriptive in nature and labelled for data review, and further analytic coding surfaced concepts and theories. After coding, key themes and patterns were organized and displayed in memos and diagrams to draw conclusions for verification by the study team. Analysis considered social and systems-level constructs that highlighted experiential meaning associated with the use ATCs in Turkana County, and specifically the use of the nREM platform for ATCs. Outcome variables included ATC strengths, user motivation and satisfaction, county-level benefits, and respondent "wishes" for future integration of the innovation into the health system.

## Results

Between January 2017 and June 2018, the nREM project facilitated 104 provider-to-provider ATCs in Turkana County. ATCs were used to: a) support NPCs to manage cases locally and either avoid or encourage referral, and b) inform and/or receive approval from physicians at the Lodwar County Referral Hospital (LCRH) to move the patient, ensuring that patient movement and care was coordinated. ATCs were initiated from 17 of the 22 intervention facilities. The majority of ATCs were between NPCs (nurses and COs) in Turkana and remote MOs (73.5%). ATCs were rarely initiated by MOs in Turkana (12.5%), and the remainder were between NPCs and remote specialists (14.1%).

Cases managed spanned a wide range of clinical fields and complexity, including obstetrics and gynaecology (e.g. varicose veins in an HIV-positive pregnant woman), paediatrics (e.g. five-year-old with missed immunizations), neurology (e.g. fever and anaemia in an epileptic boy), dermatology (e.g. septic keloids not responding to treatment), neglected tropical diseases (e.g. Brucellosis, Kala azar), and orthopaedics (e.g. dislocated hip in a 16-year-old following a fall from a camel). The youngest case managed was a three-month-old with sepsis, and the oldest an 82-year-old with pneumonia.

Appreciative Inquiry into ATC integration into the Turkana County health system suggested beneficial impact on healthcare providers, the health system, and patients/caregivers and wider Turkana society (Fig 3). Table 2 provides examples of quotes from each participant type referring to ATC benefits to these stakeholders. Among providers, ATCs increased communication and collaboration, skills and confidence, and professional satisfaction in both NPCs and remote physicians. Responses from both healthcare providers and County Health Management Team (CHMT) members in Turkana suggest that ATCs likely contributed not only to improved patient care, but also to increased equity and efficiency in the health system, data access and usage for decision-making, and efficiency of referrals. Constructs related to health quality were not restricted to health outcomes, but extended to wider societal benefits through increased equity and efficiency, and also through increased user-confidence in a health system supported by ATCs.

**Fig 3. Areas of influence of ATCs in the Turkana health system.** Findings from the 'Discovery' phase of Appreciative Inquiry revealed impact from ATCs on different stakeholders in the Turkana health system. Patient-level health outcomes from ATCs likely result from the diverse intermediary outcomes and process improvements.

### Benefits to healthcare providers

Providers reported wide-ranging benefits to themselves and their peers from engaging with ATCs, including: increased communication and collaboration, improved skills and confidence, and improved professional satisfaction.

**Improved communication and collaboration.** ATCs enabled NPCs in Turkana to access clinical support from highly-qualified physicians/specialists. It also linked them to distant referral facilities with improved communication, teamwork, and continuity of care for patients recommended for referral. Whereas previously, NPCs in rural health facilities would have little-to-no communication with a physician, either for professional development or to discuss a case, ATCs provided a way for clear and directed communication to take place. Opportunities for collaboration, teamwork, and partnership within and between health systems also increased. Prior to engaging with ATCs, COs were afraid to approach MOs, and MOs were equally shy to approach specialists. Teleconsultations contributed to reducing social status barriers between healthcare providers, with health professionals of different cadres seeing themselves as part of a team delivering quality care to patients in Turkana.

"*Even though I am alone, I feel that we are many; I am not alone.*"–Nurse, Turkana County

NPCs and remote physicians alike addressed the platform's ability to enhance a pre-existing perception of 'brotherhood' that existed among Kenyans working in remote locations. This closeness was further enhanced by a belief that healthcare professionals worked towards a

**Table 2. Sample quotes from participants on the benefits of ATCs.** Quotes from different participant types suggesting that the impacts of ATCs spanned different stakeholders–patients and the Turkana society, healthcare providers–both local and remote, and the health system in Turkana.

| Participant type | Benefits to providers | Benefits to health system | Benefits to patients and society |
|---|---|---|---|
| *Remote physicians* | "Sometimes we fall into a routine [in our medical practice] . . . but cases from Turkana are different. I'm passionate about paediatrics, so I get excited when I get a call from Health-E-Net. I get excited because it's something different. It's motivating for me." | "This platform can be used to show public health [data]; it can be used on top of the clinical [data]. The analysis can really inform public health programming because it actually gets data from far flung areas which might not be reporting so well. It can be a way of bringing out the trends in those areas." | "the population around, most of them you know are not well off, in case of any referrals they might end up selling their goats or something. . . so that they may get around. I thought that the platform is quite beneficial for them because most of them can trek 100 kilometres from where they are. . . to just see a physician and be given medication and just go back. . . [teleconsultations are] quite beneficial to the county." |
| *Non-Physician Clinicians* | "We are all healthcare professionals. It's not about names now; it's about healthcare professionals delivering services as a team." | "It cuts costs, because ambulance coming all the way. . .its very expensive in terms of fuel, you pay the driver,. . . so basically when I talk with a consultant on the other side and tells me, this is very simple, just do this. . .we have saved a lot actually. | "Without [nREM], it would have taken at least 48 hours for [patient name] to get to a specialist in Eldoret. This could have been too late considering her age. . . . I am very excited; I was able to help the girl from this facility without having to send the family away." |
| *County Stakeholders* | "The platform offers an opportunity for them to interact and discuss cases, and in the process build a cohesive, strong team that is focused on the patient." | "Even the issue of collecting information, it has been useful to us because it makes our work easier and also for. . .the people I supervise, the rest of the health workers. So even decision making, because you can only make decisions if you have quality data and information so it contributes to that. I get to have quality information, which can help me make the right decisions." | "To me, it is something that eats right into the patient; it's a patient-centred approach, this whole element [of teleconsultations]." |
| *Physicians in Referral Hospital* | "If you get a referral [on the platform], it helps you prepare. Like one of the referrals was for blood transfusion. Usually we don't have blood in the hospital, but if you know there is somebody coming with a severe anemia, it helps you be ready for the patients, so your management at least will be prompt." | "Instead of focusing all the resources in bringing such a patient to your facility, if it's a patient who can be managed at that level, you can actually advise on how the patient can be managed there and not bring the patient in the main hospital. In that sense you reduce the referrals which are usually very costly." | "I appreciated also through the [platform] the patient could be able to access better health care." "The patient I got could actually die, had we not prepared." |

common cause for the benefit of their people. From the perspective of remote physicians, the ability to speak directly to the NPC about a case, whether through the chat function on the platform or by phone, added greatly to their ability to provide sound feedback on case management.

**Improved skills and confidence.** Both NPCs and remote physicians reported a direct link between using the ATC platform and growing technical skills and confidence in their clinical practice. For NPCs, uploading challenging cases and receiving feedback and support from physicians allowed them to learn and further develop clinical skills.

*"I feel satisfied because when I enter the case, I get more knowledge because the doctors or the consultants will add more flesh to what I have, to my diagnosis, when they now send the feedback. I am able to see it and say, 'Ok, I could have done this, I could have done that.' So, it adds some more knowledge to me in my experience"*–CO, Turkana County

NPCs also saw ATCs as a capacity-building mechanism, where they have direct access to technical advice and perhaps even longer-term mentorship from senior colleagues. These interactions assisted NPCs to better interpret patient signs and symptoms under the guidance of physicians. One NPC commented that knowledge acquired through one ATC had improved his ability to handle a similar case the next time without support.

Additionally, the ATC platform's systematic approach to entering patient data using a "structured narrative" approach aligned with best practices in documenting patient history. NPCs improved their skills in case documentation through training and the structure of the platform. Both NPCs and remote physicians commented on the benefits of such organization of clinical information to improve consistency in documenting patient information by NPCs, and in aiding remote physician review of each case.

> "*It also makes me feel like now my records are well kept. . . Anything I need, especially for medical legal [purposes], I can get it from the platform.*"–NPC, Turkana County

> "*. . . the [clinicians] who have been presenting materials in the platform have very good write ups on the history, particularly the previous history of what has gone on . . . from the information I have probably 80% of what I need to make a decision.*"–Remote physician, Nairobi

The ATC platform's capacity for uploading clinical and radiology images also added relevant details to a case review, increasing a remote physician's ability to make an informed decision. Remote physicians also noted that the interaction with challenging and rare cases encouraged them to study and consult other physicians, building their own skills and strengthening profession networks in the process.

> "*I don't want to pass on the wrong information, which may affect someone else's life. It's a huge responsibility. . .and motivates me to be more knowledgeable.*"–Remote physician, Nairobi

> "*. . . as a professional I feel it's giving me an opportunity to grow . . . it also helps keep building on medical knowledge especially considering where I work now, the range of cases we see are quite different, so for me it really helps me keep my mind sharp.*"–Remote physician, Nairobi

**Increased professional satisfaction.** NPCs and remote physicians expressed pride and satisfaction in their work as a result of the nREM project. Through ATCs, NPCs believed their patients could receive the highest quality of care possible, even in remote, low-resourced settings. The satisfaction of knowing they gave patients the best care possible generated a virtuous cycle, increasing patient confidence in their ability to deliver high-quality care.

> "*My patients see me as a professional. They always come back to my health facility because they say they've been helped.*"–Nurse, Turkana County

Remote physicians also expressed professional satisfaction in their ability to review, study, and provide accurate feedback to a challenging case in rural Turkana, including feelings of usefulness and pride in serving fellow Kenyans.

## Benefits to the healthcare system

Providers and health system managers reported that ATCs facilitated greater equity and efficiency in healthcare delivery, specifically facilitating equitable access to care, improving referral mechanisms, reducing healthcare delivery cost and time, and increasing access to data for decision making.

**Increased equity in access to care.** Increasing equity in healthcare access was particularly noteworthy in a vast county like Turkana where disparities in access to healthcare are wide [32]. Vast geography, poor infrastructure, a nomadic population, and a long history of marginalisation has resulted in a poorer healthcare services and health outcomes in Turkana

compared to urban regions of Kenya [31]. Through ATCs, though, respondents expressed a great sense of enthusiasm about patients' ability to access the same quality of care as someone living in Nairobi.

> "*Patients are getting great care from all this [access to teleconsultations].*"–NPC, Turkana County

Quantitative data from the nREM project on age and sex distribution of physical referrals versus teleconsultations support the impact of ATCs on equity in access to physicians (Fig 4). ATCs tend to be more equally distributed between adult men and women, and children, when compared with physical referrals (across the county and not restricted to nREM sites) during the study period, which were dominated by adult men (difference not statistically significant).

**Increased efficiency of referral.** Access to ATCs made NPCs more likely to consider local management with remote support. Prior to the nREM project NPCs were more likely to refer challenging cases to referral hospitals. Multiple respondents spoke to the reduction in unnecessary referrals, and alleviating the pressure on referral facilities to accommodate patients who could be treated at primary level. When referrals were necessary, ATCs improved coordination of care between PHCs and the referral hospital.

> "*In the past I would have said, 'Let's just refer,' but when someone senior tells you, 'No, you know what? It's malaria, let's do this;' or 'It's meningitis, let's do this treatment;' It helps.*"– Nurse, Turkana County

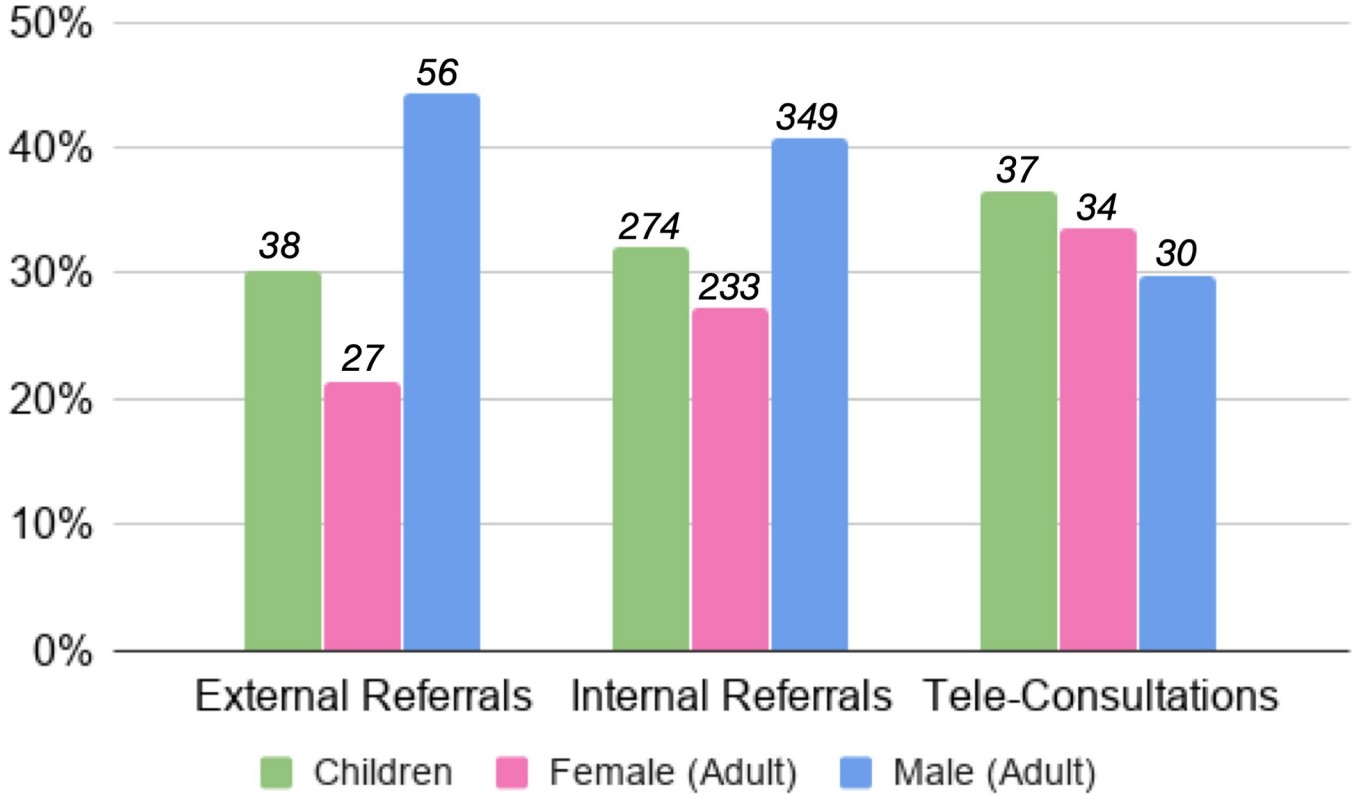

**Fig 4. Distribution of referrals and ATCs between adult men, women and children.** Numbers of external referrals (from LCRH to facilities outside Turkana County) internal referrals (from PHCs in Turkana to LCRH), and teleconsultations/electronic referrals on the ATC platform between January 2017 and June 2018. External and internal referrals involved traditional patient movement (with or without ambulances), without electronic documentation on the ATC platform. Difference not statistically significant ($\chi^2$ (4, n = 1,078) = 7.1703, p = .13).

*"[ATCs] reduce patient influx, as in referral. . .patients who are occupying beds, you know, unnecessarily. So at least when we have patient referred to county referral centre, this is a patient who is worthy, who is really in need of that service, so we reduce the pressure in the referral facility, because we are able to do them wherever we are."*–CO, Turkana County

*"Before the platform was introduced, you had to call Lodwar that you are bringing a patient with this and this condition, and sometimes you come with the patient, only to reach Lodwar and you become aware that they are not prepared to handle the case. But with the introduction of the platform, once you enter the case into the tablet, by the time the patient is referred. . .you come when they are already aware about the case of the patient and they have prepared. For those with minor surgical cases, they are attended to and patients are very happy. They come and say that this computer is actually helping us."*–CO, Turkana County

County Government officials spoke to the "millions of Kenyan shillings saved" through preventing unnecessary referrals (e.g. costs for ambulances, fuel, driver, accompanying clinicians, hospital expenses). While transportation cost savings from avoided referrals are relatively simple to calculate, savings from improvements in clinical care, patient outcomes and quality of life are harder to estimate, as are costs borne by patients and other actors outside the health system. These are however clearly articulated by clinicians and administrators in the county.

*"[recording referrals on the nREM Platform] saves them a lot of things. One, they should write referral form. Two, they should look for an ambulance, in case they need an ambulance. Three, those ambulances. . .they need fuel, so they have to inquire about fuel. They inquire about allowances for the drivers. Then even the nurse who is accompanying the patient. So, it's a big process when you are referring patients, rather than when you are using the platform, yes. I think it somehow you know reduces their works schedule."*–CHMT member, Turkana County

**Increased data access and use.**   Data collected through ATCs were available to county stakeholders for decision-making in health systems strengthening efforts. These data brought out trends in primary healthcare in remote areas, while also highlighting gaps in diagnostics, management, and personnel and equipment availability. Several county stakeholders chose to use ATC data from the nREM project for their own system-level reporting and decision-making. Availability of these data reportedly saved time that was previously used to search for information from each health facility. ATC data, available in electronic format, enabled health system managers to assess county achievements and set future milestones more easily. County stakeholders also valued the data availed through ATCs for purposes of developing health system strategies for the county, as well as advocating for the amendment and implementation of various health policies including those around facility and personnel development and the distribution of funds.

## Benefits to patients and care-givers

As described by both NPCs and remote physicians, patients and their care-givers directly benefit from improved NPC capacity to manage cases locally, improved quality of services, and improved referral management.

**Improved patient management and quality of care.**   By receiving remote support from MOs and specialists for managing cases locally, NPCs were able to offer improved clinical care to patients. While this is difficult to demonstrate quantitatively, responses from clinicians in interviews and FGDs suggest that patients received better services as a result of ATCs, at least

partly due to improved clinical capacity and confidence in the NPCs. This was also reflected in a perception of increased patient confidence in their work and the system broadly.

*"I feel connected to a wide range of information, so I'm confident about my care."*–Nurse, Turkana County

*"Patients are also confident that the health care workers are working on him or her. And in case of anything, they have confidence that we will forward their case and we will get immediate feedback on how to give them better health management."*–CO, Turkana County

**Increased social benefits to patients and communities.** The potential impact of illness on nomadic communities in Turkana County was starkly highlighted, as the entire community participates in caring for the family and livestock. Improved care processes and outcomes for patients in these nomadic communities, through ATCs, contributed to maintaining the social fabric of these societies.

*". . . with nomadic [communities] you know there [are] a lot insecurities involved and so the families tend to like to stay together . . . especially when you move men and they are the decision makers of the family, there will be a lot of things involved, there will be poor decision making, there is kind of feeling of insecurity because for them its men who provide security for the family."*–CHMT member, Turkana County

Avoiding unnecessary referrals has significant impact on the patient and the community at large. Averted referrals also reported as cost saving for the patient. Several respondents shared that members of nomadic communities sell livestock to cover the costs of sending a family member to a referral hospital for medical care. Patients often cannot afford the cost of transportation, accommodation, and workdays lost to either accompany a sick child to a referral hospital, or seek treatment him/herself. Therefore, treatment received at a PHC close to the patient contributed significantly to reducing the burden of sickness on communities.

*"Referral here is hard. You may refer a patient who may never go to where you have told them to go. . .because of their condition; there is high poverty. Some may just not go to the hospital."*–CHMT member, Turkana County

*"There is also a cost attached to clients. They have to pay, and even the people who take care of them, you see this is the cost sometimes it's not seen, but it does cost a lot, people moving where they are to another place. . .The client himself and the caretaker, that person maybe has left some work, there is also cost related to that, skipping duties, maybe working, maybe doing a business, we count that time also a cost. If that decision can be arrived at where you are and you do the right thing from there you will have saved really a lot."*–CO, Turkana County

**Improved health outcomes.** While this study did not aim to measure health outcomes at patient level, there are several cases where the clinicians using the ATC platform felt they had contributed to such outcomes, for example, by receiving instructions for managing the patient, or receiving patient information that improved referral.

*"Usually, the protocol for obtaining blood from the blood bank is tedious and time consuming. In emergencies such as [patient name], having the prior information on the platform enabled us to save her life."*–MO, Turkana County

## Discussion

This Appreciative Inquiry study, built around an implementation of a specific modality of tele-medicine–provider-to-provider (versus direct, provider-to-patient), asynchronous (versus real-time) teleconsultations using a distributed network of remote providers (versus a hub-and-spoke model)–aimed to explore the wider benefits of teleconsultations that could inform both future implementations and evaluations in low-resource settings. The study revealed diverse impacts on a variety of stakeholders–patients, providers, and the wider health system–as illustrated in Table 2 and Fig 3. Many of these potential benefits of ATCs are 'upstream' of clinical services, and there are likely multiple pathways to any health impact on patients. These non-clinical, 'intermediate' outcomes are however important to understand, design for, and evaluate in any complex health service delivery intervention such as ATCs [36, 37].

### Efficiency of referral

The study demonstrates the benefits of innovations like ATCs derived by the wider health system through gains in efficiency and equity. Efficiency gains through improved referral can be framed using the three delays model [38, 39]. ATCs in Turkana contributed to addressing delay #1 (delay in decision to seek care) and delay #3 (delay in provision of adequate care). Non-adherence to referral advice is a major barrier to LMIC health systems, resulting in late presentations and high mortality from potentially treatable conditions [40, 41]. Several clinicians spoke to the use of the ATC platform to prepare the referral hospital for the arrival of a patient (e.g. by ensuring supply of blood)–reducing delay #3. The use of ATCs to avoid unnecessary referral, emphasise the importance of referral when deemed necessary, and improve coordination of care, can not only reduce costs to the health system (and allow reallocation of scare resources), but also improve health outcomes through early detection and treatment.

### Equity in access

Most respondents raised the concept of equity in terms of the limited care available to the population of Turkana, either explicitly or implicitly comparing services there with those in other regions of Kenya. Data from the nREM project however also highlights the inequities within care delivered in Turkana–adult women are least likely to present in the referral hospital when compared with adult men or children. While the reasons for adult men being most likely to be referred could be diverse (e.g. men could present later and with more serious conditions needing referral), anecdotal evidence and interviews suggested that adult women are more likely to refuse referral, citing concerns about the care of the family while they are away. This is supported by the relatively equal distribution of men, women and children receiving ATCs or electronic referral support (while not different statistically). These data highlight another potential process outcome for telemedicine interventions–their effect on social inequities in access to healthcare.

### Patient confidence and care-seeking behaviour

Intermediate outcomes at the patient level, as perceived by NPC respondents upon reflection on patient experience, included increased confidence and trust in the local NPC and the primary healthcare system. These are likely to lead to increased care seeking at the PHC and increased compliance with referral if deemed necessary. In a widely dispersed and nomadic population, such trust combined with a reduction of unnecessary referrals (and related social disruption, displacement and costs) are important to maintain the social fabric of

communities. Attempts to measure such intangible impacts of complex interventions like ATCs are essential if the longer-term sustainability of health systems are to be understood [42].

## Provider motivation and communication

Among providers, one area that benefited greatly from ATCs in Turkana was the motivation, confidence and capacity of NPCs to deliver primary healthcare in remote, low-resource settings. Even remote, physicians reported the effect of teleconsultations on keeping them up-do-date, especially with rarely seen cases (e.g. Turkana is an endemic region for Kala azar) [43]. Such outcomes of teleconsultations on clinicians' self-perceived clinical competence have rarely been studied, whether in high-income or LMIC settings [13, 44]. The findings also provide insights into how improved communications between different cadres of health workers has the potential not only to improve healthcare services, but also to positively influence traditional, rigid hierarchies among health workers in LMICs [45]. Future teleconsultation interventions in LMIC settings could benefit greatly by considering their impact on motivation, teamwork and interpersonal interactions between health workers–elements that profoundly impact health outcomes.

## Capturing outcomes of importance

The Appreciative Inquiry framework served as an effective approach to document factors that influenced social and systems-level change. The iterative nature of Appreciative Inquiry highlighted elements that may have previously been neglected from either research inquiry or participant response; however, the method's focus on transformative change allowed the study team to "see" into social fabrics of the health system, and how ATCs influenced that system.

It is important to take such a systems approach to technology-enabled healthcare service evaluation [46]. Teleconsultations, whether in synchronous or asynchronous paradigms, link the demand for case-specific medical advice/support with scarce resources of skilled physicians/specialists. At scale, matching a distributed and diverse resource of human skills to a specific local demand that spans varied clinical situations and contexts, is likely to need complex technology combining modern mobile phone technologies with algorithms to determine the "intellectual proximity" between local need and global resource. By enabling efficient utilisation and redistribution of scarce human resources for health in LMICs, teleconsultations are likely to have widespread impact on individuals, health systems and societies. A strengths-based methodology like Appreciative Inquiry allows for sustaining such efforts by capturing positive deviants or outliers, examining reasons influencing their success, and designing to maximise their impact.

## Study limitations

Due to the technological nature of the nREM intervention and the rare integration of innovations into marginalised health systems, Turkana-based participants may have responded with enthusiasm that distorted truth due to their desire for continuity of services–a form of social desirability bias. Appreciative Inquiry's emphasis on storytelling and grounding responses in experience sought to mitigate exaggeration in self-reported data. Second, the study lead was a non-Kenyan who may have been perceived as influential in the continuation of services in Turkana. The study team attempted to minimize this bias by emphasizing the external position of the lead author, and that future growth models could only be improved with honest responses. Finally, data from stakeholders engaged in nREM prior to 2016 (in sites other than Turkana) were not collected for this study, in order to understand the influence of ATCs on

one specific health (sub)system. The lack of these data may have limited the breadth of research findings. The study team did conduct interviews with volunteer physicians who engaged in ATCs prior to 2016 to understand if and how their experience, as well as the model at that time, could add to the data analysed for this study; they determined these data as irrelevant for the question under study.

## Conclusions

As the role of physicians in LMIC health systems evolves from frontline care to incorporating support of frontline NPCs [25], there is growing need for innovation in teleconsultation processes and technologies, and their integration into routine workflows for healthcare service delivery. While the promise of telemedicine to delivery healthcare to remote, rural populations has remained unchanged [1], their practical implementation has illuminated numerous challenges, not least the (in)ability to demonstrate clinical impact [20]. While health outcomes are critical in understanding telemedicine efficacy over time, these are just one measure of improvements associated with such innovations. This study contributes toward integration of technological innovations into LMIC health systems by understanding wider, systems-level benefits of ATCs. Understanding the effect of their use in the health system in Turkana provides early insights into how they might be used and evaluated more widely.

## Acknowledgments

The authors acknowledge the support of Dr. Gilchrist Lokoel, Dr. Nelson Lolos and other members of the Turkana County Ministry of Health Services & Sanitation for their support for the nREM project and thank them for sharing their deep love and understanding of the Turkana people and culture. The authors thank Meghan Kumar for reviewing the manuscript and providing valuable insights into complex health systems research, Kelvin Waweru for research assistance in Turkana, and the Health-E-Net developer team for developing and supporting the technology platform and its users.

## Author Contributions

**Conceptualization:** M. Whitney Fry, Pratap Kumar.

**Data curation:** M. Whitney Fry, Salima Saidi, Vanessa Kithyoma.

**Formal analysis:** M. Whitney Fry.

**Funding acquisition:** Pratap Kumar.

**Investigation:** M. Whitney Fry, Abdirahman Musa, Pratap Kumar.

**Methodology:** M. Whitney Fry, Pratap Kumar.

**Project administration:** M. Whitney Fry, Salima Saidi, Abdirahman Musa, Vanessa Kithyoma, Pratap Kumar.

**Resources:** Salima Saidi.

**Supervision:** Vanessa Kithyoma, Pratap Kumar.

**Validation:** M. Whitney Fry, Salima Saidi, Abdirahman Musa, Vanessa Kithyoma.

**Visualization:** M. Whitney Fry, Salima Saidi, Pratap Kumar.

**Writing – original draft:** M. Whitney Fry, Salima Saidi.

**Writing – review & editing:** Pratap Kumar.

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
