## [Decision Letter · Decision Letter 0]

11 Mar 2020

PONE-D-20-00735

“Even though I am alone, I feel that we are many” – An appreciative inquiry study of asynchronous, provider-to-provider tele-consultations in Turkana, Kenya

PLOS ONE

Dear Dr. Kumar,

Thank you for submitting your manuscript to PLOS ONE. After careful consideration, we feel that it has merit but does not fully meet PLOS ONE’s publication criteria as it currently stands. Therefore, we invite you to submit a revised version of the manuscript that addresses the points raised during the review process by both reviewers.

We would appreciate receiving your revised manuscript by Apr 25 2020 11:59PM. Please include the following items when submitting your revised manuscript:

We look forward to receiving your revised manuscript.

Kind regards,

Abhishek Makkar, M.D.

Academic Editor

PLOS ONE

Journal Requirements:

2. Please include a copy of the interview guide used in the study, in both the original language and English, as Supporting Information, or include a citation if it has been published previously.

3. Thank you for providing the following Funding Statement: 

"The work and all authors were supported by a County Innovation Challenge Fund award from DFID (CICF-INN-R1-033) to Health-E-Net Limited. PK was also supported by a Stars in Global Health award (S5 0420-01) from Grand Challenges Canada. The funders had no role in study design, data collection and analysis, decision to publish, or preparation of the manuscript."

We note that one or more of the authors is affiliated with the funding organization, indicating the funder may have had some role in the design, data collection, analysis or preparation of your manuscript for publication; in other words, the funder played an indirect role through the participation of the co-authors.

If the funding organization did not play a role in the study design, data collection and analysis, decision to publish, or preparation of the manuscript and only provided financial support in the form of authors' salaries and/or research materials, please review your statements relating to the author contributions, and ensure you have specifically and accurately indicated the role(s) that these authors had in your study in the Author Contributions section of the online submission form. Please make any necessary amendments directly within this section of the online submission form.  Please also update your Funding Statement to include the following statement: “The funder provided support in the form of salaries for authors [insert relevant initials], but did not have any additional role in the study design, data collection and analysis, decision to publish, or preparation of the manuscript. The specific roles of these authors are articulated in the ‘author contributions’ section.”

If the funding organization did have an additional role, please state and explain that role within your Funding Statement.

Please also provide an updated Competing Interests Statement declaring this commercial affiliation along with any other relevant declarations relating to employment, consultancy, patents, products in development, or marketed products, etc.  

Reviewers' comments:

Reviewer's Responses to Questions

**Comments to the Author**

1. Is the manuscript technically sound, and do the data support the conclusions?

Reviewer #1: Partly

Reviewer #2: Partly

2. Has the statistical analysis been performed appropriately and rigorously? 

Reviewer #1: N/A

Reviewer #2: N/A

3. Have the authors made all data underlying the findings in their manuscript fully available?

Reviewer #1: No

Reviewer #2: Yes

4. Is the manuscript presented in an intelligible fashion and written in standard English?

Reviewer #1: Yes

Reviewer #2: No

5. Review Comments to the Author

Reviewer #1: This is a qualitative study using Appreciative Inquiry that studied the impact of asynchronous tele-consultations on providers and health systems in Kenya.

Overall, this is a very lengthy, verbose manuscript. The authors should critically review to make the content more concise.

Abstract:

This may read better if the authors use a more active voice, e.g. "We demonstrated that provider-to-provider asynchronous tele-consultations impacted multiple stakeholders."

The last portion of the final sentence is inaccurate as the authors did not study "patient-level health outcomes". Please rephrase to be more accurate, e.g. "...health service delivery interventions that can benefit providers and health systems."

Introduction:

The third sentence is lengthy and difficult to understand. Please rephrase.

I would recommend that the authors do a more extensive literature as some of the background is inaccurate. This includes the final sentence of the first paragraph. For example, many studies have been published on telestroke that demonstrate both its clinical and cost effectiveness. Similarly, there are publications on asynchronous telemedicine services beyond tele-dermatology, i.e. tele-radiology, tele-ophthalmology, etc.

At the end of the introduction, the authors include three objectives. It does not seem that the third objective was addressed in the body of the manuscript, i.e. how ATCs can be sustained in LMIC health systems. Please remove this objective.

Materials and Methods:

The authors state that research participants were recruited based on frequency of usage of the nREM system. This is concerning as it could introduce significant selection bias into the study. That is, by only collecting data from people who frequently used the system, they are missing data from those who did not use it. This is a major limitation as the non-users may have a very different experience with and impression of the impact of nREM on providers, patients, and the healthcare system.

Results:

Under the section "Improved Skills and Confidence", how did this system improve the technical skills if it was asynchronous? This is not intuitive. Please provide an example. It is much easier to understand how the asynchronous consults would improve knowledge and confidence -- but technical skills seems less likely.

For the quantitative data on referrals vs tele-consultations (Figure 4), please do a comparative analysis to determine whether the difference is statistically significant.

Under "Benefits to Patients and Care givers", the first sentence states that there is a "direct association with patient-level outcomes...." This is inaccurate as no patient-level outcomes are reported in this study. I would rephase to something like "patient level experience".

Discussion:

In the second paragraph on Page 26, the authors should rephrase the first sentence. There were no intermediate patient-level outcomes measured. It is more accurate to say that this is the NPCs perception of the patient experience as there was no data collected directly from the patients, e.g. via surveys, interviews, focus groups, etc.

Please remove "Akin to dating apps" in the conclusion. This has no place in medical literature.

Reviewer #2: Comments also attached. I think this is a nice study, and adds a value to in terms of provider engagement and addressing access to health care in low recourse communities. It would be great to revise the way it's written. It was a little bit hard to read at some points. I also would comment on the sustainability of the platform

Reviewer 2#

Dear Drs Fry, Saidi, Musa, Kithyoma and Kumar

Thank you for your work to ensure access to your patients regardless of their location.

 Nice and valuable work to demonstrate the value of provider engagement in low-middle income communities to deliver health care by connecting rural providers and their patients to services at distant sites and promoting patient-centered health care and equal access

Introduction section:

Standardize the terminology used:

Doctor to doctor vs provider to provider

It might be beneficial to describe the scope of practice of the NPC.

Page 9

Under MATERIALS AND METHODS section:

The way it written, it is describing Appreciative Inquiry methodology study design rather than the valuable work that the study intends to deliver. It would be great to describe the demographics of not only the subjects but the type of the consults, include/move the details of platform, training of the providers and measurements in this section. This will help the reader to understand the value of “understanding how the asynchronous tele-consultation intervention contributed to social and systems-level advances, and how these improvements can be leveraged to integrate tele-consultations”

Abbreviations should be explained within parentheses at their first mention in the manuscript. 

Page 10-11:

Would like to see a description of the data included

**Facilitation of key informant interviews (KIIs)** and focus group discussions (FGDs) was

conducted by a public health and Appreciative Inquiry specialists within the Health-E-Net

team:

Few suggestions/comments:

 Unclear if the was a standardized interview tool utilized Is there any professional relationship between the team and the interviews (volunteer consulting physicians (MOs and specialists) ?? if so, what other measures used to minimize biases?

Page 12-

Under the results questions:

 It seems that there were some open-ended questions and story telling as mentioned in the limitation section, it might be more beneficial to have standardized questions to address the feasibility, challenges and benefits Information may be difficult to quantify or organize

Page 20-21

It might be more powerful to describe the impact of ATC on patient care e.g description of avoided transfers, number of patients who were able to receive their care at their home town

6. PLOS authors have the option to publish the peer review history of their article (what does this mean?). If published, this will include your full peer review and any attached files.

Reviewer #1: No

Reviewer #2: No

---

## [Author Response · Author response to Decision Letter 0]

14 May 2020

All reviewer comments have been addressed in the attached file titled Response to reviewers.pdf

---

## [Decision Letter · Decision Letter 1]

10 Jul 2020

PONE-D-20-00735R1

"Even though I am alone, I feel that we are many" - An appreciative inquiry study of asynchronous, provider-to-provider teleconsultations in Turkana, Kenya

PLOS ONE

Dear Dr. Kumar,

Thank you for submitting your manuscript to PLOS ONE. After careful consideration, we feel that it has merit but does not fully meet PLOS ONE’s publication criteria as it currently stands. Therefore, we invite you to submit a revised version of the manuscript that addresses the points raised during the review process.

We look forward to receiving your revised manuscript.

Kind regards,

Abhishek Makkar, M.D.

Academic Editor

PLOS ONE

Additional Editor Comments (if provided):

Dear Dr. Kumar,

Thanks for submitting revised version. Based on our review, most of previous reviewer comments have been addressed. It does need minor changes to it before it can be accepted for publication. Please address reviewer comments and resubmit revised version by 8/21/20.

Reviewers' comments:

Reviewer's Responses to Questions

**Comments to the Author**

1. If the authors have adequately addressed your comments raised in a previous round of review and you feel that this manuscript is now acceptable for publication, you may indicate that here to bypass the “Comments to the Author” section, enter your conflict of interest statement in the “Confidential to Editor” section, and submit your "Accept" recommendation.

Reviewer #3: (No Response)

2. Is the manuscript technically sound, and do the data support the conclusions?

Reviewer #3: Yes

3. Has the statistical analysis been performed appropriately and rigorously? 

Reviewer #3: Yes

4. Have the authors made all data underlying the findings in their manuscript fully available?

Reviewer #3: Yes

5. Is the manuscript presented in an intelligible fashion and written in standard English?

Reviewer #3: Yes

6. Review Comments to the Author

Reviewer #3: The authors have made significant improvements by responding to each of the reviewers suggestions, by re-editing the abstract and manuscript and adding a statistical analysis as suggested.

Two issues remain:

First, on the Title, Author, Affiliation page, the authors need to revisit the Plos One formatting sample and make corrections accordingly.

Second, Reviewer 1, comment 5 suggests the authors need a more extensive literature review incorporating newer (or other) articles reporting on asynchronous telemedicine services. While it is acceptable to argue that the focus of the current article is on studies "at scale" the existence of a broader literature can still be acknowledged. I would suggest something along the lines of "while there is a growing literature on the use of asynchronous telemedicine applications to assist medical providers in underserved areas (provide several references here), the need for studies on applications at scale remains." Or something like this. In this way you can acknowledge there are studies on asynchronous applications and still justify the importance of your study by highlighting how it differs from these.

Under Materials and Methods, after Figure 1 paragraph, the paragraph beginning "a volunteer network..." Sentence 2 reads "The network included current and former employees the county..." In your edits you removed too much here. Should read "...current and former employees of the county..."

7. PLOS authors have the option to publish the peer review history of their article (what does this mean?). If published, this will include your full peer review and any attached files.

Reviewer #3: No

---

## [Author Response · Author response to Decision Letter 1]

14 Aug 2020

The responses have been submitted in the attached file titled "Response to reviewers-20200810.pdf"

---

## [Editor Report · Decision Letter 2]

25 Aug 2020

"Even though I am alone, I feel that we are many" - An appreciative inquiry study of asynchronous, provider-to-provider teleconsultations in Turkana, Kenya

PONE-D-20-00735R2

Dear Dr. Kumar,

We’re pleased to inform you that your manuscript has been judged scientifically suitable for publication and will be formally accepted for publication once it meets all outstanding technical requirements.

Kind regards,

Abhishek Makkar, M.D.

Academic Editor

PLOS ONE

---

## [Editor Report · Acceptance letter]

4 Sep 2020

PONE-D-20-00735R2 

"Even though I am alone, I feel that we are many" - An appreciative inquiry study of asynchronous, provider-to-provider teleconsultations in Turkana, Kenya 

Dear Dr. Kumar:

I'm pleased to inform you that your manuscript has been deemed suitable for publication in PLOS ONE. Congratulations! Your manuscript is now with our production department. 

Kind regards, 

on behalf of

Dr. Abhishek Makkar 

Academic Editor

PLOS ONE